# Insights into the dynamic trajectories of protein filament division revealed by numerical investigation into the mathematical model of pure fragmentation

**Magali Tournus** [1] *, **Miguel Escobedo** [2], **Wei-Feng Xue** [3,4], **Marie Doumic** [4]

**1** Centrale Marseille, I2M, UMR 7373, CNRS, Aix-Marseille université, Marseille, France, **2** Universidad del País Vasco, Departamento de Matemáticas, Bilbao, Spain, **3** School of Biosciences, University of Kent, Canterbury, United Kingdom, **4** INRIA Rocquencourt, équipe-projet BANG, domaine de Voluceau, Rocquencourt, France

☯ These authors contributed equally to this work.
* mtournus@math.cnrs.fr

**Data Availability Statement:** All relevant data are within the manuscript and its Supporting information files. The code is available on GitHub,

## Abstract

The dynamics by which polymeric protein filaments divide in the presence of negligible growth, for example due to the depletion of free monomeric precursors, can be described by the universal mathematical equations of 'pure fragmentation'. The rates of fragmentation reactions reflect the stability of the protein filaments towards breakage, which is of importance in biology and biomedicine for instance in governing the creation of amyloid seeds and the propagation of prions. Here, we devised from mathematical theory inversion formulae to recover the division rates and division kernel information from time-dependent experimental measurements of filament size distribution. The numerical approach to systematically analyze the behaviour of pure fragmentation trajectories was also developed. We illustrate how these formulae can be used, provide some insights on their robustness, and show how they inform the design of experiments to measure fibril fragmentation dynamics. These advances are made possible by our central theoretical result on how the length distribution profile of the solution to the pure fragmentation equation aligns with a steady distribution profile for large times.

## Author summary

Amyloid fibrils are fibrillar protein structures involved in many neurodegenerative illnesses, such as Parkinson's disease or Alzheimer's disease. To propagate in disease, these misfolded protein aggregates must grow and divide to proliferate. Therefore, the intrinsic characteristics of their division, including the division rate and the pattern of division in terms of whether the fibrils are likely to break in the middle or at the edges, impact the disease aetiology. Here, we discovered mathematical formulae that can be used to directly extract the fibril division characteristics from recent experiments data obtained from time-dependent fibril length distribution measurements. We explain how these formulae

via the following link https://github.com/mtournus/Fragmentation.

**Funding:** M.E. is supported by DGES Grant MTM2014-52347-C2-1-R and Basque Government Grant IT641-13. M. D. and M.T. were supported by the ERC Starting Grant SKIPPERAD (number 306321). W.-F.X. was supported by funding from Inria, and Biotechnology and Biological Sciences Research Council (BBSRC) UK grants BB/J008001/1 and BB/S003312/1. The funders had no role in study design, data collection and analysis, decision to publish, or preparation of the manuscript.

**Competing interests:** The authors have declared that no competing interests exist.

can be used, and prove the robustness of the division rate formula where small errors in the measurement leads to small errors in the division rate. We also demonstrate that the mathematical formula is not robust enough to precisely decipher the pattern of division in the data, and suggest instead new future experimental design with short time measurements in experiments starting with fibril suspensions where all fibrils have similar size, which would be suitable to provide improved estimates.

## Introduction

How can we extract information on the stability and dynamics of proteins nano-filaments from population distribution data? This general question is of topical interest due to the ever-increasing evidence to suggest that the fragmentation of amyloid and prion protein fibrils [1] are associated with their biological response ranging from being inert, functional to toxic, infectious and pathological [2]. The experimental methods to characterize the dynamics of amyloid fibril fragmentation has been evolving from indirect bulk kinetics measurements [3] to direct observations in population level time-dependent nano-imaging experiments ([4, 5]). To analyze the division of protein filaments when the experimental information we have is at the level of the population distribution, for instance when the type of data we currently can acquire are time-point samples of fibril length distributions and individual dividing particles cannot yet be isolated and tracked, the pure fragmentation equation reveals to be a powerful mathematical tool. The pure fragmentation equation describes the time evolution of a population of fibril particles structured by their size $x$ that divide into smaller particles. The underlying assumption is that the dimensions of each particle govern its division dynamics: each particle of length $x$ is assumed to divide with a rate $B(x)$, and when a particle of size $y$ divides, it gives rise to a particle of size $x$ with a probability encoded in the fragmentation kernel $\kappa(x, y)$. Though the fragmentation equation describes the dynamics at the level of the whole population, the properties $B$ and $\kappa$ have a natural interpretation in terms of the microscopic stability of the polymers. In this report, we address the question of determining the parameters $B$ and $\kappa$ from the size distribution of the protein filament suspension at different times.

The application of the pure fragmentation equation can be traced back to almost 100 years. In the seminal paper by Kolmogorov [6], a fragmentation model for grinding particles was proposed. The model is discrete with respect to time but continuous in the structuring variable corresponding to the size of the particle. This allowed Kolmogorov to work with explicit formulae. The unknown property in the Kolmogorov model is the cumulative distribution function of the particle sizes, and he assumed a constant fragmentation rate and a generic kernel preventing the creation of too many small particles. Under these assumptions, he obtains that the cumulative distribution follows, asymptotically in time, a log-normal distribution. At the very end of the paper, Kolmogorov suggests that his study should be extended to generic fragmentation rates, and especially the ones with a power law dependence on particle size, i.e.

$$B(x) = \alpha x^{\gamma}. \tag{1}$$

In parallel, Montroll and Simha [7] developed a discrete model for pure fragmentation of long-chain molecules such as starch with the restrictions that the kernel follows a uniform distribution (each bond has the same probability of fission), and only fission into two parts is allowed (compared to Kolmogorov's model that allows the fission into $n$ particles). In the late 70's [8], the problem was again considered for the purpose of studying the degradation of long chains under high shear mechanical action. This was encouraged by new techniques to obtain

measurements of the so-called "molecular weight distribution" in a closed system with constant total mass. This was the first time a dependence of the fragmentation rate on the size of particles is studied. The authors suggested that for the molecular system they studied, if the power $\gamma = 1$, then mechanistically, the fragmentation kernel should be uniform. For $\gamma < 1$, which is the value they obtained ($\gamma = 2/3$), they suggested that the bonds on the edges of the molecules are more reactive than those in the centre. In this case, the fragmentation equations were solved numerically, and the fragmentation rate was determined from the average length of molecules, based on an approximation valid for a monodisperse suspension (we detail this approximation in S1 Appendix). The other parameters ($\alpha$ and $\kappa$) were obtained by fitting their model to the evolution of the total number of molecules. To estimate the fragmentation kernel, they considered three different types of kernels and suggested that the best-fit kernel was one described by a parabolic function, although the selection criteria were not detailed. In a theoretical paper by Ballauff and Wolf [9], the same discrete model was studied and three fragmentation kernels were considered: a uniform kernel, a Gaussian kernel, and a Dirac kernel where particles can only split exactly at their centre. An example of the time-dependent solution is plotted in each case, however, again, the overall criterion of kernel selection is through simulations, with no precise objective protocol suggested. A series of theoretical works by McGrady and Ziff followed in the 1980's, focusing exclusively on analytical formulae of the continuous model. In [10], they provided fundamental solutions that involve hypergeometric functions for a uniform kernel, and for a monomial fragmentation rate with $\gamma = 2/m$ with $m \in \mathbb{Z}$. In [11], they provided explicit formulae of the fundamental solution of the pure fragmentation equation with a uniform kernel and monomial fragmentation rate for any $\gamma$, a uniform kernel in the case where particles break into 3 pieces instead of 2, as well as for $\gamma = 3$ combined with a parabolic kernel centered at the particle centre, justified by the parabolicity of the Poiseuille flow. Typically, their solution is made of a sum of two terms, one term where the initial condition vanishes exponentially, and the other term where the profile of a stationary state arises. Using these explicit solutions, they noticed, just like Kolmogorov did, that a stationary distribution shape profile arises asymptotically after rescaling. From the 1970's onward, size structured population models were extensively developed by mathematicians for biological applications, (see [12]). The particles under consideration were bacterial and non-bacterial cells, microtubules, etc. For these systems, the 'particles' undergo division as well as growth, which led to the development and application of growth-fragmentation equations. From the 1990's, a large set of mathematical studies were focused on the division equations and related models [13], in particular on the long-time behaviours [14–16]. To deal with the major issue of model calibration, mathematicians also developed theories to recover some parameters, for instance [17, 18] where the authors determined a robust estimate of the division rate of bacterial cells from noisy measurements of the size distribution profiles of the cells at the end of the experiments, and the time evolution of the total number of cells, see also [19] and the references therein. More recently, a theory was developed [20] to estimate both the division rate and the division kernel from the measurement of the particle distribution profile at the end of the experiment, under assumptions on the division rate being given by the simple power law $\alpha x^\gamma$. Another approach emerging to estimate the division kernel is the use of stochastic individual based models by studying the underlying stochastic branching processes [21].

While the universality of the fragmentation equation is demonstrated in its applicability ranging from physical processes such as the grinding of rocks, to chemical processes such as the degradation of long chain starch molecules and biological processes such as cell division, the application we exemplify here is the mechanistic laws governing the division and propagation of filamentous amyloid structures. These proteinaceous fibrils can be associated with human diseases such as Alzheimer's disease, Parkinson's disease [22], type 2 diabetes, prion

diseases and systemic amyloidosis. The fragmentation of amyloid fibrils has been shown to enhance their cytotoxic potential by generating large numbers of small active particles [23]. Likewise, the fragmentation of prion particles that are transmissible amyloid results in an increase in their infective potential [24]. Recently, as a proof of concept, we reported a new experimental approach [5] where the stability towards breakage under mechanical perturbation for different types of amyloid fibrils were analyzed and quantitatively compared. We determined the division rates and the type of fragmentation kernels associated to each type of amyloid fibrils. These data suggested that the proteins that are involved in diseases may be overall less stable toward breakage and generate larger numbers of small active particles than their non-disease associated counterparts. In the context of the experimental data presented in [5], and as pointed out in [9], the experimental context may have a considerable impact on the loci at which the fibril is more likely to break up. Therefore, it is important to develop a general method based on a common mathematical platform, which can be applied to the analysis and comparison of experimental data from a wide range of amyloid systems and conditions.

In this report we provide a detailed explanation of the mathematical method based on the analysis of the pure fragmentation equation used in [5], together with a thorough numerical investigation of the influence of the three key parameters of the model, and the numerical algorithm used to estimate the fragmentation rate and kernel from experimental measurements. We focus on the case of 'pure fragmentation' of amyloid protein fibrils, i.e. on experiments where other growth reactions such as nucleation, polymerization and/or coagulation could be neglected. We also do not consider nonlinear fragmentation reactions, which may be induced by collisions or interactions between fibril particles, since in our context the fibril particles can be considered dilute so that this effect may be neglected. We provide inversion formulae to recover the three parameters $\gamma$, $\alpha$ and $\kappa$ from experimental measurements of the particle length distribution at different times where samples of fibril lengths are taken but no information on the total number of particles or the total mass of the suspension is directly available. In particular, our method does not rely on finding the best-fit of model distributions to the data or on the goodness-of-fit comparison between models. Instead we demonstrate robust analytical inversion formulae that express the parameters as functions that can be directly computed from the solution of the equation. The method and the analysis presented here are general and can be useful in other contexts. But importantly, the mathematical results will inform the design of experiments tailored to evaluate and compare the dynamical stabilities of protein filaments.

## Theory

In this section, we summarize the mathematical results that are the theoretical foundation of our method.

### The pure fragmentation model

We consider a population of amyloid protein fibrils, which are filamentous and pseudo linear particles, undergoing a process of 'pure fragmentation', where the only phenomenon taken into account is the division of any parent particle into two daughter particles. In this case, the rates of growth processes such as nucleation, polymerization, coagulation etc. are considered to be negligible in the experiments, for example due to the lack of monomeric precursors. The modelling assumptions we make on the fragmentation process are as follows.

*Assumption 1*: the fragmentation rate depends only upon the size of the parent particle undergoing division, and follows a power law, namely, the first order rate constant of particles of size $x$ breaking into two pieces is $B(x) = \alpha x^\gamma$ for some $\alpha > 0$. We also impose $\gamma > 0$, which

means that larger particles are more likely to break up than small particles. This assumption is necessary for the asymptotic behaviour (4) to happen. For $\gamma = 0$, no self-similar behaviour occurs [16], whereas for $\gamma < 0$, shattering (sometimes referred to as 'dust formation') occurs in finite time [25].

*Assumption 2*: the fragmentation reaction is self-similar, meaning that the sites of fragmentation on particles are invariant with size rescaling, that is the site of fragmentation on a particle can be described as a ratio between the position and the total length of the particle.

The fragmentation kernel $\kappa$ is a property that describes the probability distribution of the length of the daughter particles formed in each fragmentation event, assuming that such a fragmentation event takes place. Assumption 1 is justified since the fragmentation reaction considered in the experiments is promoted due to a single type of perturbation, in the case of [5] mechanical in nature. The particles in the sample suspension are also homogeneous in terms of being formed by the same monomer precursors and only differ by their size. In particular, the fragmentation rate is considered to be independent of the history of each particle, and on the fate of other particles. Assumption 2 is justified by the fact that the fragmentation behaviour for rods follow the scaling pattern as discussed in [26].

As amyloid protein fibrils are supramolecular polymer structures, the fibril particles considered here are made of monomeric units. There are two main approaches to describe the evolution of the fibril population. The population of particles can be described by the number of particles $u_\ell(t)$ composed with $\ell$ monomers at time $t$,

$$\begin{cases} u_\ell'(t) = -\alpha(\ell r)^\gamma u_\ell(t) + \alpha \sum_{j=k+1}^{+\infty} \frac{1}{jr} \kappa\left(\frac{\ell}{j}\right)(jr)^\gamma u_\ell(t), \ \ t > 0, \ \ell = 0 \ldots N, \\ u_\ell(0) = u_\ell^0, \ \ \ell = 0 \ldots N, \end{cases}$$ (2)

where $r$ is the average length of one monomer. We refer to Eq (2) as the discrete model. Alternatively, when the number of monomers composing each particle is assumed to be sufficiently large, we can write a continuous version of the model. The unknown is the density $u(t, x)$ of particles of length $x$ at time $t$ and the model is written as follows:

$$\begin{cases} \frac{\partial u}{\partial t}(t, x) = -\alpha x^\gamma u(t, x) + \alpha \int_x^\infty \frac{1}{y} \kappa\left(\frac{x}{y}\right) y^\gamma u(t, y) dy, \ \ t > 0, \ x \geq 0, \\ u(0, x) = u^0(x), \ \ x > 0. \end{cases}$$ (3)

The advantage of the discrete framework is its validity even when the number of monomers in the particles is small, which could be the case at very long time scales for fragmentation experiments. As for the continuous framework, the main advantage it that it is mathematically convenient since some explicit formulae exist for some specific parameters, and it enables partial differential analysis results to be used to understand the qualitative behaviour of the system. The behaviour of these two models should not differ in the time of the experiments we analyse. Therefore for our analysis, we focus on the continuous framework. For the solutions to (3), mass conservation (i.e. $\int x u(t, x) dx$ does not depend on time) is guaranteed by the condition $\int z \kappa(z) dz = 1$. We also assume that the fibrils can only divide into two, i.e. we impose $\int \kappa(z) dz = 2$ (no ternary break-up).

**An inversion formula for $\gamma$: Dynamics of the moments.** The long time behaviour of the solutions to (3) is now well-known by mathematicians [14]: the solution converges after rescaling to some steady profile $g$ in the sense that

$$t^{-2/\gamma} u(t, t^{-1/\gamma} x) \underset{t \to \infty}{\longrightarrow} g(x),$$ (4)

and where $g$ is the solution of

$$yg'(y) + (2 + \alpha\gamma y^\gamma)g(y) = \alpha\gamma \int_y^\infty \frac{1}{v}\kappa\left(\frac{y}{v}\right)v^\gamma g(v) \ dv, \quad \int_0^\infty yg(y)dy = \rho, \tag{5}$$

where $\rho$ only depends on the initial condition $u^0$ through $\rho = \int xu^0(x)dx$. In imaging experiments, we sample lengths of particles present in the population at each time point. Therefore, we introduce the measured quantity:

$$f(t, x) = \frac{u(t, x)}{\int_0^\infty u(t, x)dx}. \tag{6}$$

We define the moment of order $q$ of the distribution $f$ as

$$M_q(t) = \int_0^\infty x^q f(t, x)dx. \tag{7}$$

We deduce directly from (4) that

$$t^{-1/\gamma}f\left(t, t^{-1/\gamma}x\right) \underset{t\to\infty}{\longrightarrow} \frac{g(x)}{\int_0^\infty g(y)dy}. \tag{8}$$

A space integration of the above formula gives us

$$\log M_q(t) = -\frac{q}{\gamma}\log(t) + C(q), \quad t \text{ large}, \tag{9}$$

for the constant $C(q) = \log\left(\frac{\int_0^\infty g(y)y^q dy}{\int_0^\infty g(y)dy}\right)$. In particular, the first moment ($q = 1$), being the average length, can be evaluated directly from the length measurements. This provides us with a method to extract $\gamma$ from the data because the log-log dynamics of the mass tends to a straight line whose slope is equal to $-1/\gamma$, provided that the regime with steady distribution shape profile has been reached. Notice that $C(0) = 1$ and $C(1) = \frac{\rho}{\int_0^\infty g(y)dy}$. Importantly, the asymptotic straight line depends on the parameters of the model (e.g. its slope depends on $\gamma$, and its position depends on $\gamma$, $\alpha$, $\kappa$ through $g$) but not on the initial length distribution.

Eq (9) shows elegantly that, when applied with $q = 1$, the number average molecular weight (proportional to the average length of fibrils) $M_1(t)$ decays linearly for large times independently of the initial length distribution when plotted on a loglog scale. We refer to this characteristic line as the asymptotic line. At shorter time scales, $M_1$ is also decaying.

We note that our method to recover $\gamma$ works even if the particles can break up into more than two particles, indeed Eq (9) does not use the information of the number of particles produced by each breakage. The authors of [8] also use the dynamics of the moments to estimate $\gamma$, in the case of breakage of dextran molecules through acid hydrolysis. However, the approach in [8] is a special case with an assumption of monodispersity, and a model selection approach comparing different solutions with different $\gamma$ values was used (see a full comparison detailed in S1 Appendix).

## The Mellin transform

The inversion formula for $\alpha$ and $\kappa$ strongly relies on the Mellin transform, which appears to be an intrinsic feature of the pure fragmentation equation. For any function (or generalized function) over $\mathbb{R}^+$, we recall that the Mellin transform $\mathcal{M}[\mu]$ of $\mu$ is defined through the integral

$$\mathcal{M}[\mu](s) = \int_0^{+\infty} x^{s-1} \ d\mu(x), \tag{10}$$

for those values of $s$ in the complex plane for which the integral exists. We define for $\Re e(s) > 1$ $G(s) := \mathcal{M}[g](s)$ and $K(s) := \mathcal{M}[\kappa](s)$. The Mellin transform turns the differential Eq (5) into the following non-local functional equation:

$$(2 - s)G(s) = \alpha\gamma(K(s) - 1)G(s + \gamma) \ \forall s \in \mathbb{C}, \ \Re e(s) > 1. \tag{11}$$

**An inversion formulae for $\alpha$ and $\kappa$.** Since the fission is only binary, $K(1) = 2$. Thus, using the Mellin transform, we obtain (see [20] for the mathematical justification)

$$\alpha = \frac{G(1)}{\gamma G(1 + \gamma)} = \frac{\rho}{G(1 + \gamma)}. \tag{12}$$

We emphasize that, contrarily to $\gamma$, the estimate on $\alpha$ mainly relies on the binary fission assumption.

Estimating the division kernel $\kappa$ reveals a much harder and more ill-posed problem compared to that of $\gamma$ and $\alpha$. Once $\alpha$ and $\gamma$ are known, we may formally divide Eq (11) by $G(s + \gamma)$ and obtain

$$K(s) = 1 + \frac{(2 - s)G(s)}{\alpha\gamma G(s + \gamma)}. \tag{13}$$

The properties of the kernel $\kappa$ are such that the inverse Mellin transform of $K$ is well defined and equal to $\kappa$ (see for instance [27](Theorem 11.10.1). Therefore the fragmentation kernel $\kappa$ is given by the inverse Mellin transform of $1 + \frac{(2-s)G(s)}{\alpha\gamma G(s+\gamma)}$, provided that the (complex valued) denominator does not vanish. In fact, it is mathematically proved in [20] that there exists $s_0 > 2$ such that the denominator $G(s + \gamma)$ does not vanish. Then, for this specific $s_0$ we have

$$\kappa(z) = \frac{1}{2i\pi} \int\limits_{\Re e(s) = s_0} z^{-s}\left(1 + \frac{(2 - s)G(s)}{\alpha\gamma G(s + \gamma)}\right)ds. \tag{14}$$

The detailed mathematical justifications and proofs of the formulae given here can be found in [20]. The main idea underlying the method is the central following theoretical result: the length distribution profile of the solution to the pure fragmentation equation aligns with a steady shape for large times, and all the moments of the profile decay predicatively on an asymptotic line in log-log space. See Box 1 for a summary of the theory.

---

### Box 1: Summary of the theory

- Inversion formula for $\gamma$: $\gamma$ is obtained using Eq (9) as $\gamma = -\frac{1}{S}$, where $S$ is the slope of the straight line representing the first moment (e.g. average length) as a function of time, in log-log scale. The curve under question is a straight line for large time points.

- Inversion formula for $\alpha$: $\alpha$ is obtained using Eq (12), where $G$ is the Mellin transform of the steady shape of the length distribution for large times.

- Inversion formula for $\kappa$: $\kappa$ is obtained using formula (14) together with $\gamma$ and $\alpha$. Again, $G$ is the Mellin transform of the steady shape of the length distribution for large times.

## Results and discussion

### Exploration of trajectories

In this section, we give an overview of the influence of the parameters on the stationary profile of the self-similar length distribution and on its transient behaviour.

*Influence of $\gamma$.* It is proven in the theoretical paper [28] that the parameter $\gamma$ impacts the stationary profile for large $x$, and more specifically that $g(x)$ behaves like $C \exp^{-x^\gamma/\gamma}$ as $x \to \infty$ for some $C > 0$. This property cannot be used to extract the parameter from the stationary profile $g$, since it would require to have precise experimental information for large sizes. This property is illustrated in S1(A) Fig, where the stationary profile corresponding to different values of $\gamma$ and for a gaussian kernel is plotted. For larger values of $\gamma$, since decay at larger particle sizes is faster, the stationary profile is more concentrated around $x = 0$ (the integral of $xg(x)$ is equal to 1) compared to smaller $\gamma$ values. The role of $\gamma$ on the overall shape of $g$ is highly non-linear, and for all other parameters fixed, the overall shape can vary with $\gamma$. This is illustrated on Fig 1A for $\alpha = 1$ and the specific kernel $\kappa$ displayed in the inset, the stationary profile has different qualitative behaviours for $\gamma = 0.8, 1, 1.5$ and 2. The influence of $\gamma$ on the time evolution of the length distribution $f$ is described by Formula (9). The moments of order $z$ of the profile $f$ (for example its moment of order 1: the average size of fibrils) decrease linearly with time at log-log scale. Depending on the initial moments, the evolution of the moments can have two different shapes as illustrated on Fig 2. For example, the average length $M_1(t)$ can stay completely below the asymptotic line as illustrated by the green line, or above the asymptotic line as illustrated by the blue line. See Fig 1B, for an illustration of the trajectories with simulated data starting from different initial average lengths.

*Influence of $\alpha$.* If for the initial data $u^0(x)$, the solution to the fragmentation equation for $\alpha = 1$ is $u(t, x)$, then the solution to the fragmentation equation for the same initial data, the same values for $\gamma$ and $\kappa$, and $\alpha > 0$ is $u_\alpha(t, x) = \frac{1}{\alpha} u(\alpha t, x)$. Further if the stationary state for $\alpha = 1$ is $g$, then, for $\alpha > 0$, it is $g_\alpha(y) = \alpha^{2/\gamma-1} g(\alpha^{1/\gamma} y)$. Indeed then,

$$t^{-2/\gamma} u(t, t^{-1/\gamma}x) \to g(x), \tag{15}$$

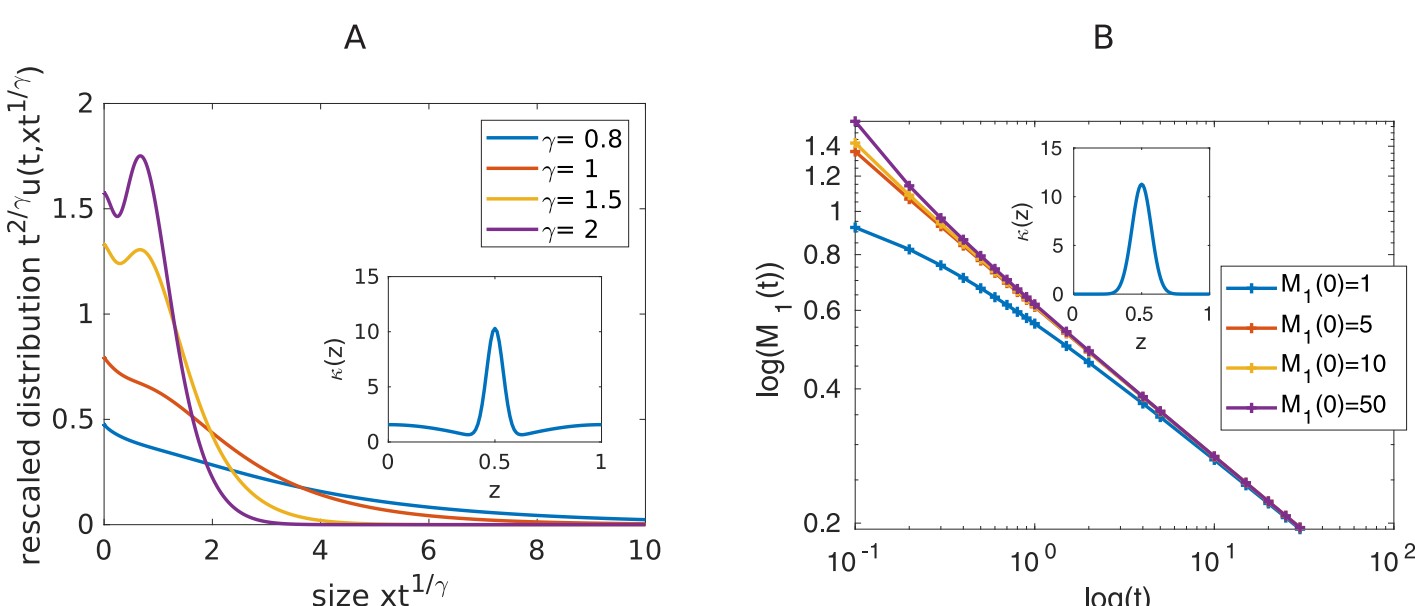

**Fig 1. Influence of $\gamma$ on the stationary length distribution profile, and transient dynamics of the average length.** A: Stationary profile for different values of $\gamma$ and $\alpha = 1$, $t_f = 200$. B: Time evolution of the mass $M_1(t)$ in a log-log scale. The initial conditions are spread gaussian with different masses 1, 10, 50 or 100 and $\gamma = 1$.

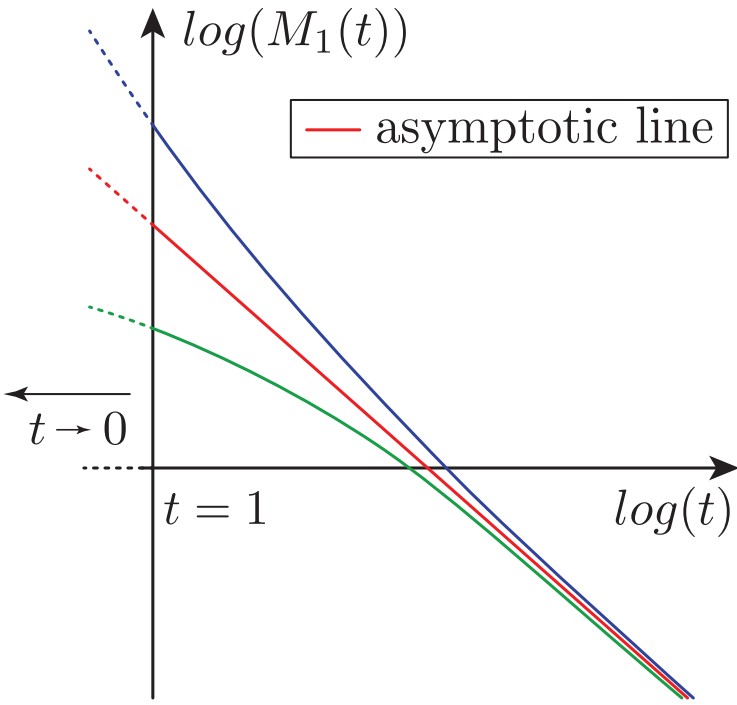

**Fig 2. Illustration of two possible scenarii for $t > 1$ depending on the initial moment of the system.** The first moment $M_1(t)$ (the distribution mean) can stay below (green) or above the asymptotic line (blue). Both behaviours have been observed numerically. In all cases, the moment $M_1(t)$ is decreasing with time and aligns to the asymptotic line (straight line in red) for large time.

then, setting $\tau = \alpha t$

$$t^{-2/\gamma} u_\alpha(t, t^{-1/\gamma} x) = \frac{t^{-2/\gamma}}{\alpha} u(\alpha t, t^{-1/\gamma} x) = \alpha^{2/\gamma-1} \tau^{-2/\gamma} u(\tau, \tau^{-1/\gamma} \alpha^{1/\gamma} x) \longrightarrow g_\alpha(x). \qquad (16)$$

This is illustrated on Fig 3A. We conclude that the parameter $\alpha$ acts as a time scaling term. This property cannot be used to recover $\alpha$, since from experiment we only know $g$ up to a multiplicative factor.

*Influence of the kernel $\kappa$.* We first explored the influence of $\kappa$ on the stationary profile $g$. In first approximation, smooth kernels can be classified into two classes: Within class A, the kernels are such that $\kappa(0) = \kappa(1) = 0$, and within class B, kernels are such that $\kappa(0) > 0$ and $\kappa(1) > 0$. On Fig 3B the stationary profiles for a selection of six different kernels are displayed. As seen, on Fig 3, right, whether the kernel belongs to class A or B can be read directly on the shape of the stationary profile. For kernels of class A, the stationary profile is zero at $x = 0$ and is unimodal (one peak), and for kernels of class B, the stationary profile is non-zero at $x = 0$ and decreasing in a neighborhood of 0. This is consistent with the theoretical results of [28] which state that if $k(z) \sim C_\kappa z^\epsilon$ for $C_\kappa > 0$ and $\epsilon > -1$ around $z = 0$, then $g(x) \sim C_g x^\epsilon$ for some constant $C_g > 0$. However, within a given class, it is difficult to extract the shape of the kernel from the mere information of the stationary profile. In particular, within class A, using only the stationary information, it is not possible to distinguish one peak kernels from two peaked kernels, nor distinguish between Gaussian with small or large spread (see Fig 3B and S1(C) Fig). S1 Fig shows the stationary profiles for kernels that have features resembling both classes. For kernels of class A, we only observe stationary profiles with one peak. On S1(B) Fig, we show stationary profiles corresponding to a kernel composed of the sum of two Gaussian

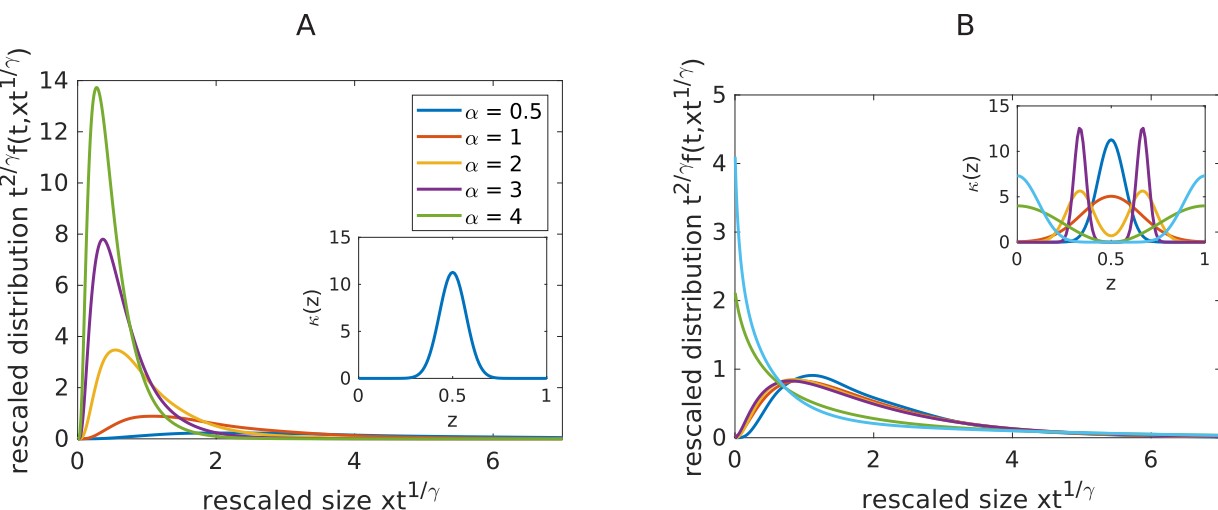

**Fig 3. Influence of the parameter $\alpha$ and $\kappa$ on the stationary length distribution profile of the fragmentation equation.** A: Stationary profiles for different values of $\alpha$, $\gamma = 1$ and $t_f = 50$. The kernel $\kappa$ used is plotted on the inset. B: Stationary profiles for different kernels $\kappa$ (the kernels are displayed on the inset Figure on the top right). Parameters: $\gamma = \alpha = 1$, $t_f = 50$.

functions. The closer to the boundary the two Gaussian functions are (and the more the kernel resembles one of class B), the closer to the boundary the peak of $g$ is as well (curve in yellow). On S1(C) Fig, it is seen that for kernels within class B, provided that $\kappa(0)$ is large enough, having some mass around $z = 1/2$ does not change the qualitative overall decreasing shape of the profile. However, if $\kappa(0) = \kappa(1)$ is too small, the stationary profile can be different, see S2 Fig, where a transition occurs around $\kappa(0) = 1.6$ where the shape of $g$ switches from a decreasing profile to a one-peaked profile. Nevertheless, class A kernel can be distinguished from class B kernel by the fact that the stationary distribution profile satisfies $g(0) > 0$ for class A kernels.

Next, we explored the influence of the kernel on the time evolution of the length distribution. The moments of the size distribution decay with a slope of $-1/\gamma$ on a log-log plot independently of the kernel $\kappa$, and the location of the asymptotic line is hardly dependent on kappa (Eq (9)) (see S3(A) Fig). To investigate the differences in the evolution of the length distribution from class A vs class B kernels, we applied a statistical test approach. We set the null hypothesis $\mathbf{H_0}$: "The distributions $f_a(t, .)$ and $f_b(t, .)$ respectively obtained with $\gamma = \alpha = 1$ and the two distinct kernels $\kappa_a$ and $\kappa_b$ are identical" as detailed in the Methods section. On S4 Fig, we plot the time evolution of the $p$-value for the null hypothesis, using two randomly generated samples of size $N = 200$ distributed along $f_a$ and $f_b$, evaluated using the Kolmogorov-Smirnov test. A high $p$-value indicates that the null hypothesis $\mathbf{H_0}$ cannot be rejected, which in turn means that whether the size distribution evolves by kernel $\kappa_a$ or $\kappa_b$ cannot be distinguished using the knowledge of the size distribution at time $t$. In particular, at time $t = 0$, since the size distributions are perfectly identical and equal to the initial condition, the $p$-value is equal to 1. For the pairs of kernels tested (Fig 4 and S4 Fig), the conclusion is that there may exist a time-window where two kernels result in a maximal difference in length distributions right after initial time. For example, on Fig 4, we show that when the two kernels belong to the two different classes (Fig 4A), the $p$-value is approaching zero after some long time, demonstrating that whether $\kappa$ belongs to class A or class B can be estimated by the asymptotic behaviour described by $g$. On the contrary, when the two kernels belong to the same class (Fig 4 and S4 Fig), the $p$-value is large for large times and depends on the initial condition for early time. Thus, in the case of comparing and estimating the precise fragmentation kernel within a class, the

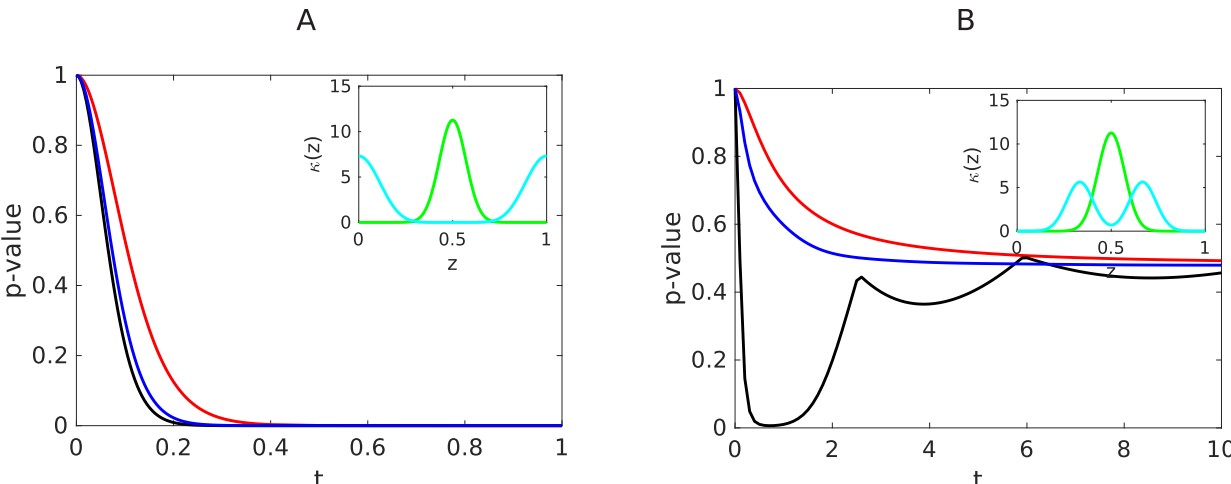

**Fig 4. How different are the evolutions of the two length distributions starting from the same initial condition and for $\gamma = \alpha = 1$, but resulting from two different fragmentation kernels $\kappa$?** The $p$-value associated with the null comparison hypothesis $H_0$ stating that the two length distributions are statistically identical for all time (see Methods- Statistical tests for a detailed description of the $p$-value) is used to quantify the statistical discrepancy between the two length distributions. We plot here the time evolution of the $p$-value, for 3 different initial conditions. Initial conditions: a peaked Gaussian (black), a spread Gaussian (blue), a decreasing exponential (red). A high $p$-value indicates that whether the size distribution evolves by kernel $\kappa_a$ or $\kappa_b$ cannot be distinguished using the knowledge of the size distribution at time $t$. On the contrary, a small $p$-value indicates that the size distribution corresponding to the two kernels $\kappa_a$ or $\kappa_b$ are clearly different. A: the kernels belong to two different classes. B: the kernels are in the same class.

asymptotic steady profile $g$ cannot be used. Instead, early pre-asymptotic length distributions may contain more detailed information on $\kappa$ in comparison.

## Inverse problem

In this part, we detail how we use the inversion formulae detailed in the Theory section to recover the parameters $\alpha$, $\gamma$ and $\kappa$ from measurements. First, the parameter $\gamma$ is extracted from the data using Formula (9). On Fig 5A, we plot the time evolution of the average length of the system in a log-log scale, for a Gaussian kernel, and for several different values of $\gamma$. As described by formula (9), as time goes by, each curve tends to become a straight line of slope $-1/\gamma$ on the log-log plot (also see S3(A) Fig). In particular, it is shown that the slopes of the time evolution of moments does not depend on the fragmentation kernel, even for early time points. Interestingly, this shows that we cannot reduce the model and that the size distributions are needed if $\alpha$ and the kernel $\kappa$ are to be extracted, i.e. measurements of moments are not enough for full description of the dynamical trajectories. The overall shape of the curves (see Fig 5) justifies that we can use the following protocol to determine $\gamma$. We assume that the measurements are given at time $t_i$, and we define for $i \in [1, n]$ the shape of the mass $M_e$ that we try to fit with

$$M_e(t_i; \gamma, C, t_e) = \begin{cases} Ct_e^{-1/\gamma}, & t_i \leq t_e, \\ Ct_i^{-1/\gamma}, & t_i \geq t_e, \end{cases} \tag{17}$$

where $t_e$ is the time at which the asymptotic line is considered reached i.e. the rescaled distribution $x \rightarrow t^{-1/\gamma} f(t, t^{-1\gamma} x)$ has aligned with the steady profile $g$, and $C$ is a constant. We introduce the quadratic distance between the moments $(M_1(t_i)_{i \in [1,n]})$ of order 1 we get from the experiments (the average lengths) and the theoretical moments $M_e(t_i)$ as $E(\gamma, C, t_e) = \Sigma_{i \in [1,n]} (M_e(t_i; \gamma, C, t_e) - M_1(t_i))^2$, and we define $\gamma$ as the point at which the

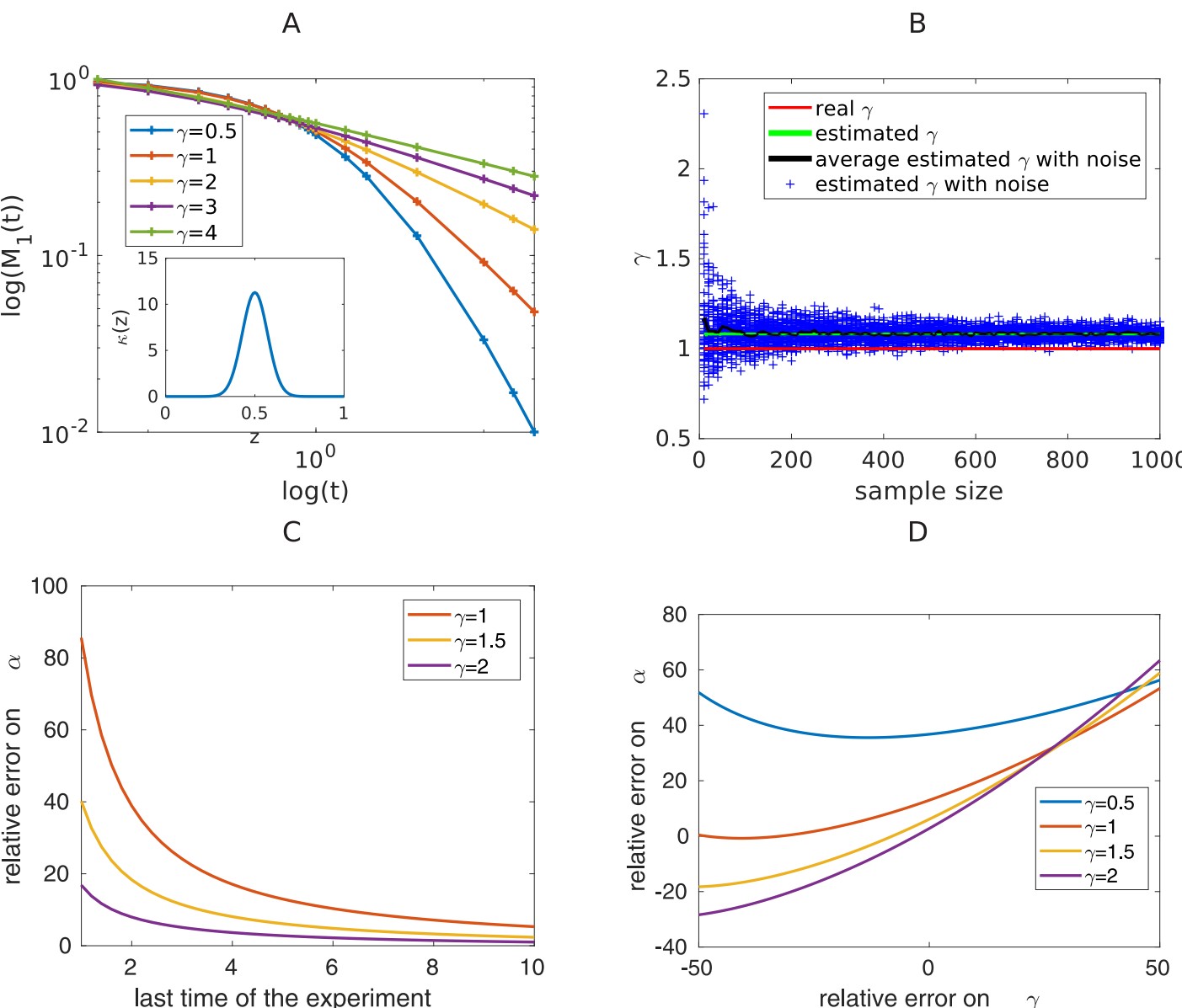

**Fig 5. Estimation of $\gamma$.** A: Time evolution of $M_1[f(t, .)]$ in a log-log scale for different values of $\gamma$. The initial condition is a spread gaussian (see Methods). B: Estimation of $\gamma$ on simulated data (ordinate) from a sample of $N$ particles (abscissa), where $N$ goes from 10 to 1000. Each experiment is repeated 50 times. The red line is the real value of $\gamma$, the green line is the estimate of $\gamma$ from the complete distribution (no sample), the blue crosses are the 50 different estimated values for $\gamma$ from 50 samples of size $N$, and for each $N$, the black line is the averaged estimated value for $\gamma$ over the 50 experiments. The time points considered are [5, 10, 15, 20]. C: Relative error on $\alpha$ (%) as a function of the last time of experiment. D: Relative error on $\alpha$ (%) as a function of the relative error on $\gamma$ (%). Here, the last time of experiment is $t = 5$.

minimum of $E$ is reached. The main advantage of this method is that it does not require any information on whether the asymptotic line is reached or not at the time points where we get measurements. S3 Fig shows a plot of the estimate of the equilibrium time given by the minimization problem. Since the protocol to recover $\gamma$ relies on the large time behaviour of the system, it is expected that the more data points for large time, the more reliable the $\gamma$ estimate is. We quantified this on Fig 5C: we define the following relative error on the estimated values $\gamma_e$ and $\alpha_e$ for $\gamma$ and $\alpha$ as $E(\gamma) = \frac{|\gamma-\gamma_e|}{\gamma}$ and $E(\alpha) = \frac{|\alpha-\alpha_e|}{\gamma}$, and we plot the error $E(\gamma)$ as a function of the last time point of the data set. We emphasize that the concavity of the moment of order 1

with respect to time in log-log scale implies that we always overestimate the value of $\gamma$. The error decreases as more time points are taken into account (see S3(D) Fig). For real data with noise, however, since particles become smaller and smaller for long times, the precision in the data is expected to increase with time until a certain limit from which the error starts to grow again. Indeed, experimentally, small particles are harder to detect, and become invisible below a threshold.

In real experiments, the measurement produces noisy data. Different kinds of noise can be distinguished. One type of noise comes from the uncertainty that is intrinsically due to the measurement devices and data processing methods. For length measurements, this type of noise is usually negligible compared to the sampling noise due to the fact that limited sample size is obtained to form the estimate of length distributions. We explore the effect of the size of the sample on the determination of $\gamma$ in Fig 5B. Our observation is that our method is robust with respect to sampling noise in the sense that for the parameters considered, with a sample of size 200 particles, the estimate for $\gamma$ (between 1 and 1.2) is correct up to 10% compared to the estimated value for $\gamma$ (which is 1.1) from the complete size distribution (no sampling). This is because the estimate of $\gamma$ is based on the evolution of the moment (e.g. average length) of the system which is a quantity that smoothen the noise being an integral. Hence, the overestimation of $\gamma$ linked to the concavity of the curve is in the same order of magnitude as the error linked to the sampling. Equipped with an estimate for $\gamma$, we estimate $g(x)$ from $u(t, x)$, using the very last data point $t_f$ namely $g(x) = t_f^{-1/\gamma} f(t_f, x t_f^{-1/\gamma})$, and then we can estimate $\alpha$ by Formula (12). As expected, the further in time we have data, the better the accuracy on the estimation of $\alpha$ (see Fig 5C). For small $\gamma$ (e.g. $\gamma = 0.5$), the error made on $\alpha$ is very large (see Fig 5C). This is due to the fact that the stationary profile is reached faster for large values of $\gamma$ under the initial condition used. The dependence of $\alpha$ on $\gamma$ is highly non linear, as well as co-dependent and we explored the effect of the error made on $\gamma$ estimate on the error made on $\alpha$ estimate. The results are plotted on Fig 5D. For fixed $\gamma$, the relative error on $\alpha$ evolves more than linearly.

## Experimental design

How to choose the initial condition and the times of measurement to acquire experimental data that can be used to optimally decipher the dynamics of particle division?

To determine $\gamma$ one needs several data points for large time. Whether the experiment has progressed long enough so that the asymptotic behaviour can be considered to be reached can be seen on the shape of the time evolution of the average length in log-log scale. As already mentioned in the previous sections, it should be a straight line in a log-log plot. To determine $\alpha$, one needs minimum the length distribution for one time point at a large time where the asymptotic line is reached. We recall here that the determination of $\alpha$ directly follows from the fact that proteins can only break into two pieces at a time. As for the kernel $\kappa$, our conclusion is that it has negligible influence on the determination of $\gamma$ and $\alpha$. To determine the class of $\kappa$, one can use the same experimental length distribution as for $\alpha$. However, to estimate the precise shape of kappa, one needs, in addition, length distribution time points close to the beginning of the experiment. To the contrary to $\gamma$ and $\alpha$ estimations that are not influenced by the initial length distribution, the best type of initial distribution to determine $\kappa$ is a highly peaked Gaussian, which corresponds to a 'monodisperse' suspension. However, it may be challenging to obtain such samples due to the physical nature of the assembly [29].

## Data analysis

We detail here how to use the theory and the Matlab code to estimate the parameters from a real experiment. First, the user should provide a set of $n$ measurements of the size distribution

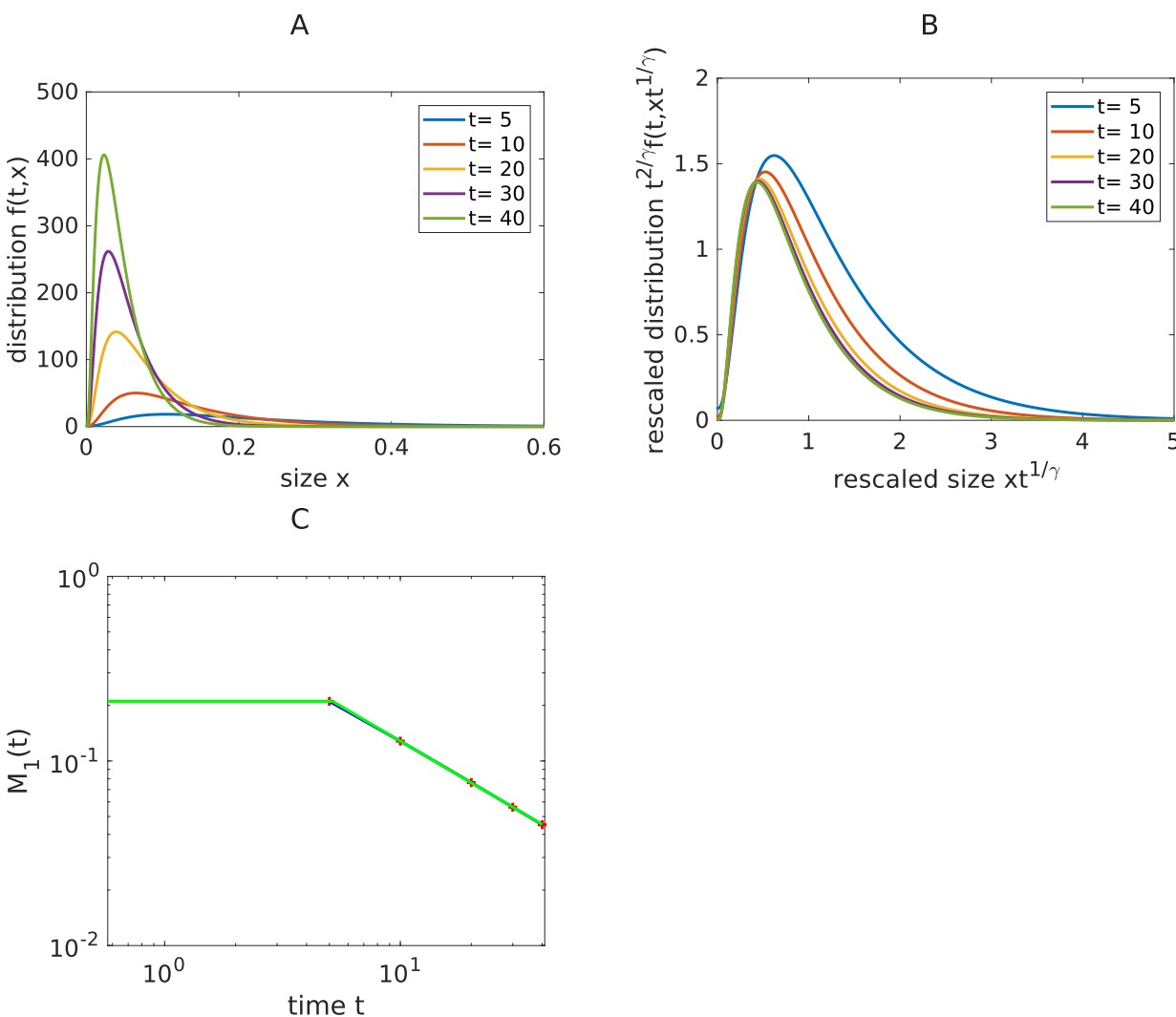

**Fig 6. Illustration of the protocol—Example 1.** A: Illustration of an example of size distribution profiles for $\gamma = 1.3$, $\alpha = 1$, and $\kappa$ is a two-peaked Gaussian kernel and the initial condition is a decreasing exponential. B: To visualize the profiles more precisely, we rescale the profiles using the real value for $\gamma$ (i.e $\gamma = 1.3$) and formula (3). What is plotted is for each time $t$ the function $t^{-1/\gamma} f(t, t^{-1\gamma} x)$, where $f(t, .)$ is the distribution profile at time $t$. C: Time evolution of the mass $M_1$ (the data points are red crosses, the solid line blue curve being a linear interpolation), compared to the estimated value $\gamma_e$ of $\gamma$ (solid line in green of slope $-1/\gamma_e$). $\gamma_e = 1.33$, $\alpha_e = 1.07$ and $T_e = 5.2$.

in the suspension at different times $t_j$. See Fig 6A and 6B for an example of such data for $t_j = 5$, 10, 20, 30, 40. We also refer to S5(A) and S5(B) Fig for an example of such data for $t_j = 0, 1, 2$, 5, 8, 13, 18 and obtained numerically with the same parameters except for the initial condition. In this last example, the experimental size distribution is a probability distribution that usually consists in a sum a dirac masses (the size of the sample at initial time is here $N = 200$), that we turn into its best fit density distribution, using here ksdensity (MatLab command). In both cases, we can observe that the proportion of small particles increases in the suspension. To have a better visualization of the profile, we plot on Fig 6B and S5(B) Fig the size distribution using the rescaling (4). For S5 Fig we observe that starting from $t = 5$, the distribution has converged to an equilibrium profile. For Fig 6, the code provides us with $\gamma_{est} = 1.33$ as an estimate for $\gamma$ and with $\alpha_{est} = 1.07$ as an estimate for $\alpha$, and the estimated equilibrium time is here $T_e = 5.2$. We display the plot of the mass evolution in a log-log scale (see Fig 6C) of the

suspension and in S5 Fig bottom panel. The estimate of $\gamma_e$ is obtained as the estimated slope of the last part (large times) of the solid line in blue.

## Methods

### Numerical simulations

To develop, explore and test a new protocol to extract the division law properties $\gamma$, $\alpha$ and $\kappa$ from experimental data based on our inversion formulae shown in the Theory section, we proceeded as follows. From an initial distribution profile, and given a set of parameters $\gamma$, $\alpha$ and $\kappa$, we created simulated data of size distributions $u$ using a numerical scheme, which is described in the next paragraph and implemented in Matlab. The three initial conditions we considered are a gaussian centered at $x = 1$ with standard deviation $\sigma = 10^{-2}$ that we refer to as the peaked gaussian, a gaussian centered at $x = 1$ with standard deviation $\sigma = 1$ that we refer to as the spread gaussian, and a decreasing exponential $f_0(x) = \exp(-x)$.

We observed numerically that the scheme is converging, and we validated it on test cases of known analytical formulae (Table 1). Subsequently, we first used the numerical scheme to explore the role of the $\gamma$, $\alpha$ and $\kappa$ parameters on the behaviour of the system. This provided insights on how and when (whether at early or late times) each parameter affects the trajectory of the system. Then, we use the distribution $f$ of the simulated data obtained with the scheme to test our method to estimate the $\gamma$, $\alpha$ and $\kappa$ parameters by comparing the estimated parameters with their known values used to generate the simulated datasets. We also added noise to the simulated data to see how it affects our estimates. We underline that the method of estimating $\gamma$, $\alpha$ and $\kappa$ based on our analytical inversion formulae is only valid in the cases where the experiments under consideration is well described by the model (3). As already mentioned, discrete models (2) are extensively used in the literature [4]. Up to a certain point, the same conclusions should also hold true for the discrete model. We employed the discrete model for comparison, see S6 Fig for a comparison between the solution of the discrete code with theoretical solutions. The good matching between the results given by the codes that discretize both the continuous and discrete models validate each of them.

Throughout this report, the time scale we consider is unitless, as $t$ represents the time in seconds divided by $t_{ref} = 1$ s.

**Table 1. Some analytical solutions of the pure fragmentation equation.** The symbol M is the Kummer's confluent hypergeometric function. The parameter $s$ is positive.

| $\gamma$ | $\alpha$ | $\kappa(z)$ | Initial data | Solution | Ref |
|---|---|---|---|---|---|
| 1 | 1 | uniform | $f_0(x)$ | $e^{-tx}\left(f_0(x) + \int_x^\infty f_0(y)(2t + t^2(y - x))dy\right)$ | [10] |
| 2 | 1 | uniform | $f_0(x)$ | $e^{-tx^2}\left(f_0(x) + \int_x^\infty 2f_0(y)ty\,dy\right)$ | [10] |
| > 0 | 1 | uniform | $f_0(x)$ | $f_0(x)e^{-tx^\gamma} + 2t\int_0^\infty y^{\gamma-1}e^{-ty^\gamma}f_0(y)\ \mathbf{M}\left[\frac{\gamma+2}{\gamma}, 2t, (y^\gamma - x^\gamma)\right]\ dy$ | [11] |
| > 0 | 1 | uniform | $e^{-sx^\gamma}$ | $e^{-(t+s)x^\gamma}\left(1 + \frac{t}{s}\right)^{2/\gamma}$ | [10] |
| 2 | 1 | uniform | $e^{-x}$ | $e^{-tx^2-x}(1 + 2t(1 + x))$ | [10] |
| 3 | $\frac{1}{6}$ | $z(1 - z)$ | $f_0(x)$ | $f_0(x)e^{-tx^3/6} + 2t\int_{y=x}^\infty e^{-ty^3/6}\int_{\ell=y}^\infty \ell f_0(\ell)d\ell\frac{dy}{y^2}$ | [11] |
| 0 | 1 | dirac in $\frac{1}{2}$ | $f_0(x)$ | $f(t, x) = e^{-t}\sum_{k=0}^\infty \frac{(4t)^k}{k!}f_0(2^k x).$ | [32] |

## Numerical scheme

We detail here the numerical scheme we use to solve (3) to generate simulated distributions and trajectories. Our method is based on [30]. We set $w = \log(x)$ and instead of directly writing a scheme on $f$, we simulate the evolution of the quantity $n(t, w) := e^{2w} u(t, e^w)$ which satisfies for $t > 0$ and $w \in \mathbb{R}$

$$\frac{\partial n}{\partial t}(t, w) = \alpha e^{\gamma w}\left(-n(t, w) + \int_0^\infty \kappa(e^{-y})e^{\gamma y}e^{-2y}n(t, y + w)dy\right), \quad n(0, w) = e^{2w}u(0, e^w). \quad (18)$$

The advantage of using a scheme on the variable $n(t, w)$ instead of $u(t, x)$ is that the quantity $n$ satisfies the conservation property

$$\frac{d}{dt}\int_{\mathbb{R}^+} n(t, w)dz = 0. \quad (19)$$

We discretize the time axis with a uniform time step $\Delta t$. For the $w$ variable, we consider a uniform grid $[w_1, \ldots, w_p, \ldots, w_I]$ of step $\Delta w$ (which corresponds to an exponential grid for $x$), with $w_p = 0$. We denote by $n_i^k$ the approximated values of the variable $n$ at time $k\Delta t$ and at $w_i = (i - p)\Delta w$. Let us observe that $w_i + w_j = (i + j - 2p)\Delta w = w_{i+j-p}$. We set the initial data

$$n_i^0 = e^{2w_i}u(0, z^{w_i}) \text{ for } i \in [1, I], \quad (20)$$

and the iteration process for $i \in [1, I], k \geq 0$,

$$n_i^{k+1} = n_i^k - \alpha\Delta t e^{\gamma w_i}n_i^{k+1} + \alpha\Delta t e^{\gamma w_i}\sum_{j=0}^{\min\{I-i, I-p\}} e^{-2w_{p+j}}\kappa(e^{-w_{p+j}})e^{\gamma w_{p+j}}n_{i+j}^k, \quad (21)$$

which is for $i \in [1, I], k \geq 0$

$$n_i^{k+1} = \frac{1}{1 + \alpha\Delta t e^{\gamma w_i}}\left(n_i^k + \alpha\Delta t e^{\gamma w_i}\sum_{j=0}^{\min\{I-i, I-p\}} e^{-2w_{p+j}}\kappa(e^{-w_{p+j}})e^{\gamma w_{p+j}}n_{i+j}^k\right). \quad (22)$$

**Remark 1** *We use an implicit scheme instead of the explicit scheme*

$$n_i^{k+1} = n_i^k - \alpha\Delta t e^{\gamma w_i}n_i^k + \alpha\Delta t e^{\gamma w_i}\sum_{j=0}^{\min\{I-i, I-p\}} e^{-2w_{p+j}}\kappa(e^{-w_{p+j}})e^{\gamma w_{p+j}}n_{i+j}^k \quad (23)$$

*since for the explicit formulation, the CFL stability condition ([31]) that guarantees positivity of the solution imposes the following upper bound on $\Delta t$*

$$\Delta t \leq \frac{1}{\alpha\exp(\gamma w_I)}. \quad (24)$$

*In some cases, for instance for real data, the CFL stability condition leads to impose $\Delta t \leq 0.01$ whereas the final time is 1 million. On the contrary, the implicit version (22) of the scheme is stable with no stability condition on $\Delta t$ and allows us to take larger values for $\Delta t$.*

An alternative numerical scheme that uses the discrete modeling approach (2) based on [4] is also used for comparison and for validating the above numerical scheme. Explicit solutions for (3) are summarized in Table 1. We use these explicit solutions to validate our numerical scheme.

## Statistical tests

We detail here the statistical test described and used in the Results and discussion section.

At each time point $t$ of the experiment, we test the null hypothesis **H$_0$** "**H$_0$**: Starting with a fixed initial distribution, the samples $f_a$ and $f_b$ respectively obtained with $\gamma = \alpha = 1$ and the two different kernel $\kappa_a$ and $\kappa_b$, have the same distribution".

Given two samples $f_a$ and $f_b$ of respective size $N_a$ and $N_b$, we define the distance

$$d_{ab} = \sup_x |F_a(x) - F_b(x)|, \tag{25}$$

where $F_a$ and $F_b$ are the empirical cumulative distribution functions associated with the samples $u_a$ and $u_b$. The Kolmogorov-Smirnov test works as follows: the $H_0$ null hypothesis is said to be rejected at the significance level $\ell$ if

$$d_{ab}^2 > -\frac{1}{2}\ln(\ell)\frac{N_a + N_b}{N_a N_b}. \tag{26}$$

Note that in the literature, the level of significance is denoted by $\alpha$ instead of $\ell$. The symbol $\alpha$ being already used for the fragmentation rate, we decided to denote the significance level by $\ell$. If the above condition is satisfied, the Kolmogorov-Smirnov test recommends not to reject the **H$_0$** hypothesis. We recall that no conclusion can be drawn if the reverse inequality is satisfied (in particular, we can never say that **H$_0$** can be statistically rejected, see [33] for a complete theory on statistical tests).

The $p$-value associated with a statistical test is the level $\ell_{lim}$ from which we consider that we cannot statistically reject the null hypothesis. The $p$-value is then a non-linear function of this distance $d_{ab}$ expressed as

$$p_{value} = \exp\left(-2\frac{N_a + N_b}{N_a N_b}d_{ab}^2\right). \tag{27}$$

What is done in general is building an estimate of the cumulative functions $F_a$ and $F_b$ using an interpolation of two samples of size $N_a$ and $N_b$. (e.g. S7 Fig). In our case, we use the exact $N_a$ and $N_b$ to compute $d_{ab}$. Let us also mention that in our context, in the case where the hypotheses **H$_0$** cannot be rejected, it means one kernel cannot be distinguished from the other using only a measurement of size $N$ at the time $t$.

## Conclusions

In this study, we presented the mathematical analysis of the pure fragmentation equation. Based on the theoretical analysis, inversion formulae to directly recover information regarding division rates $\alpha$ and $\gamma$ parameters, and division kernel $\kappa$ from time dependent experimental measurements of filament size distribution are derived. These inversion formulae allow analysis of the dynamical trajectories of fibril fragmentation without goodness of fit analysis of models. This is the basis of an analytical method that enables the systematic comparison of the stability towards division for amyloid filament of different types. We believe extracting and comparing the rates and the kernel describing fragmentation reactions reflect the stability of the protein filaments towards breakage, which is of importance in amyloid seed production and the propagation of the amyloid state in functional and disease-associated amyloid.

Here, our conclusions are that the stationary length distribution profile depends non-linearly on $\gamma$ and $\kappa$. The parameter $\gamma$ can be estimated using the measurement of two or more late-time length distribution profiles. The parameter $\alpha$ is a scaling parameter that can be estimated from one late-time length distribution profile combined with the estimated value for $\gamma$. Our

inversion formulae for the parameters $\gamma$ and $\alpha$ are proved to be robust with respect to sampling noise. We also provide an algorithm (code written in Matlab) that take as an input the measured length distribution profiles at different times and give to the user, as an output, the estimated values for $\gamma$ and $\alpha$ corresponding to the measured dynamics.

As for smooth fragmentation kernels $\kappa$, we show that they can be separated into two groups: the kernels such as $\kappa(0) = \kappa(1) = 0$, (e.g. a Gaussian function), that lead to a unimodal stationary length distribution profile, and the kernels such that $\kappa(0)$ and $\kappa(1)$ are large enough, that lead to a decreasing stationary length distribution profile around 0. However, non-trivial combinations between these two rough types of kernels may lead to highly non-trivial stationary distribution profiles. Despite these two rough classes of kernels, our work demonstrates that the knowledge of late-time length distribution profiles is not enough to identify the precise fragmentation kernel. In particular, if the kernel is a Gaussian function, its spread cannot be deduced from late-time measurements. Instead, early length distributions contain more detailed information on $\kappa$. This suggests that the experiments that can provide the best data to estimate $\gamma$ and $\alpha$ are long-time experiments starting with any initial distribution. In this case, length distributions at several time points are needed after the asymptotic regime is reached to ensure good estimates of $\gamma$ and $\alpha$. On the contrary, to estimate the fragmentation kernel $\kappa$, the experiment should rather start with a highly peaked distribution, fibrils of similar length, and the evolution of the sample length distributions should be measured at short time points before asymptotic regime is reached. Such an initial distribution is very complicated to obtain experimentally, and we explore in a future work how the spread of the initial distribution affects the estimate of $\kappa$. Such experiments are challenging to perform, and future work revealing how the spread of the initial distribution affects the estimate of $\kappa$ is also needed. A practical consideration for the experimentalist is to determine whether or when the asymptotic regime has been reached. This a theoretically challenging question, but a practical protocol (as follows) can be used to inform the design of experiments. Firstly, run a simulation of the fragmentation experiment (Matlab code is made available, see Methods) using the initial distribution that can be experimentally determined, and a first guess for the fragmentation rate and kernel parameters $\gamma$, $\alpha$ and $\kappa$. Secondly, estimate the time $T_e$ after which the curve $(\log(t), \log(M_1(t)))$ has become a line. Thirdly: perform the simulation until time $5T_e$.

Finally, we emphasize that what is assumed in the present paper is that the parameters $\gamma$, $\alpha$ and $\kappa$ are intrinsic and independent characteristics of each and every individual types of amyloid fibrils. Then, an appropriate experiment to estimate $\gamma$, $\alpha$ and $\kappa$ is one that observes the population of fibrils of one given type in the absence of growth, for example using dilute samples with depleted free monomers. It may be of interest to also estimate the intrinsic growth rate of the fibrils. The protocol we suggest is to separate the growth experiment from the fragmentation experiment, and to first estimate the fragmentation characteristics as presented in this paper, and then focus on estimating the growth rate separately. Fragmentation equations are used in many different applications, for which our method can apply to. In particular, with a modern experimental approach that would provide time-dependent size distribution profiles, the fragmentation rate and kernel can be obtained for polymers of any type [8].

## Supporting information

**S1 Appendix. Recovering $\gamma$ from the data: Comparison with [8].**
(PDF)

**S1 Fig. Influence of the parameters $\gamma$ and $\kappa$ on the stationary length distribution profile.**
(PDF)

**S2 Fig. Stationary profile for different values of $\gamma$ and $\alpha = 1$.**
(PDF)

**S3 Fig. Additional figures.**
(PDF)

**S4 Fig. Plot of the time evolution of the $p$-value corresponding to the null comparison hypothesis $H_0$, for 3 different initial conditions.**
(PDF)

**S5 Fig. Illustration of the protocol—Example 2.**
(PDF)

**S6 Fig. Convergence of the numerical scheme.**
(PDF)

**S7 Fig. Plot of the time evolution of the p-value corresponding to the Kolmogorov-Smirnov test for the $H_0$ hypothesis.**
(PDF)

## Author Contributions

**Conceptualization:** Magali Tournus, Miguel Escobedo, Wei-Feng Xue, Marie Doumic.

**Formal analysis:** Magali Tournus.

**Investigation:** Magali Tournus, Miguel Escobedo, Wei-Feng Xue, Marie Doumic.

**Supervision:** Wei-Feng Xue, Marie Doumic.

**Validation:** Wei-Feng Xue, Marie Doumic.

**Writing – original draft:** Magali Tournus.

**Writing – review & editing:** Magali Tournus, Miguel Escobedo, Wei-Feng Xue, Marie Doumic.

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
