## [Decision Letter · Decision Letter 0]

17 Feb 2021

Dear Dr Tournus,

Thank you very much for submitting your manuscript "Insights into the dynamic trajectories of protein filament division

revealed by numerical investigation into the mathematical model

of pure fragmentation" for consideration at PLOS Computational Biology. As with all papers reviewed by the journal, your manuscript was reviewed by members of the editorial board and by several independent reviewers. The reviewers appreciated the attention to an important topic. Based on the reviews, we are likely to accept this manuscript for publication, providing that you modify the manuscript according to the review recommendations.

Sincerely,

Philip K Maini

Associate Editor

PLOS Computational Biology

Mark Alber

Deputy Editor

PLOS Computational Biology

[LINK]

Reviewer's Responses to Questions

**Comments to the Authors:**

Reviewer #1: The paper by Tournus et al provides a very nice study of the dynamics of fragmenting protein filament structures. In particular, the authors have obtained explicit inversion formulas that can be used to extract key information about the rates of protein filament fragmentation in a direct way. The authors also illustrate the power of their method on numerical data and discuss how their results can inform experimental design. Overall, I think that this is a very nice study. I really enjoyed reading this paper, which is well written. The topic is also of great interest due to the relevance of protein filament fragmentation to a range of diseases, particularly in the context of infectious prions. The paper also makes a very nice contribution which is both of theoretical and experimental interest. I therefore recommend this paper for publication, subject to some minor comments outlined here below.

1) The inversion formula for gamma is an asymptotic statement valid for large times. Can the author provide some estimates for the range of times where this formula holds? In other words, how long does one have to measure the length distribution in order to see a straight line of slope q/gamma and extract gamma? Does this estimate depend on the choice of the moment, i.e. q?

2) The authors have focussed on the pure fragmentation kernel. While I am not objecting this choice, I would be very interested in knowing how their results would change in the presence of other aggregation processes, particularly growth? The mathematics in this case would be much more complex because growth introduces new couplings in the aggregation kernel. Would their methods work in this situation as well? Or can the authors suggest some alternative methods/strategies that one may use in this case?

Some typos:

1) Page 10, line 8: Should E(alpha) be related to |alpha-\\alpha_e| rather than |alpha-\\gamma_e|?

Reviewer #2: The paper is concerned with pure fragmentation processes as it occurs for polymeric protein filaments and more precisely on the inverse problem. The question is to recover the division rate and division kernel from the observation of the dynamics of filament size distribution. Being given the experimental set up, the continuous fragmentation model is used. To recover the division rate, the theory is built on the, by now, well-known self-similar profile achieved in long times. The method to build the division kernel is based on the Mellin transform according to a theory developed earlier, in particular by the authors. The shape of kappa, influences in a direct and remarkable way the shape of the asymptotic self-similar profile.

Departing findings are that neither the stationary state nor the knowledge of the mere moments are sufficient to recover the kernel kappa. An inversion procedure has to be devised which is performed here and allows to recover the fibrils distribution dynamics. A remarkable conclusion is that the kernels can be two different types (A or B here). The paper is careful about real experimental data and taking into account uncertainties in measurements and the challenge of diminishing filament size along time.

After the theoretical background, the paper describes inverse problem procedure, the needed experimental data and their analysis. It uses a numerical code validated in comparison with another code and vs analytical solutions. A statistical test is used to possibly distinguish the expected kernels.

The problem of understanding the size distribution is important for several areas as cell size distribution and fibrils dynamics for neuro-degenerative diseases. Based on recent discoveries and methods from mathematical analysis of the relevant models, the present contribution is an important step in the concrete determination of the model parameters.

Minor remarks

p6: scission —> Fission ?

p7: there is no need of this long discussion to discover that alpha is a time scale. This is obvious from (3).

Sometimes it is written “on Figure” or “in Figure”. The former being more correct.

After (13) change coma to dot.

(17) check the inequalities (how can mass change after the steady state is reached) and the space next line

p12: n cannot be both the number of measurements and the generic times

p14: these explicit solutionS

Bibliography needs some uniformisation; in [33] critIcal

Reviewer #3: This paper analyzed the ability of a previously published fragmentation model to fit real experimental data by developing several new algorithms, including a numerical method and an application of statistical hypothesis testing. The authors provide a "type" of sensitivity analysis (see my first comment below) and describe a new numerical algorithm to estimate the fragmentation rate and kernel from experimental measurements. They also derive inversion formulas to recover the three key parameters from experimental measurements. The results of this study are novel and important because the focus is on performing a practical inverse problem with respect to what can actually be experimentally measured, namely the particle length distribution at various time points in a fragmentation experiment. Another important result is that their method doesn't rely on traditional parameter estimation methods, but instead an analytic inversion formula is derived that expresses the three key model parameters as functions that can be directly computed from the solution to the fragmentation equation. Given the practical and experiment focused nature of these results, it is likely that this work will enable the design of future experiments whose purpose is to investigate the stability of protein filaments and the ensuing fragmentation process. Before publication, I suggest two items be addressed:

1) The "sensitivity analysis" presented is not rigorous (or what would typically be expected when an applied math paper refers to "sensitivity analysis") and instead the results presented are, as stated in the paper, a numerical investigation of the influence of the three key parameters. For example, a rigorous "local" sensitivity analysis would include the computation of the partial derivative of the output, i.e., the rescaled distribution, with respect to the parameter of interest, evaluated at some chosen parameter value. A "global" sensitivity analysis would include something like Morris screening or latin hypercube sampling to compute sensitivity over an entire parameter domain. Instead, the authors are changing the value of gamma and alpha and plotting the rescaled distribution at these values. My suggestion is to rename the analysis to reflect that the authors are exploring how the solutions change as they vary parameters over a chosen domain, since the name "sensitivity analysis" is widely used in the applied math literature to be either local or global types mentioned above, and this may be confusing to the reader. Alternatively, the authors could perform either a local or global sensitivity analysis and include the results.

2) While the introduction is very well written, and the results and methods include very thorough details, the conclusions section could be expanded to include a discussion or examples for the two points mentioned at the end of the introduction. Namely, (1) that "the method and the analysis presented here are general and can be useful in other contexts" and (2) "the mathematical results will inform the design of experiments tailored to evaluate and compare the dynamical stabilities of protein filaments".

**Have all data underlying the figures and results presented in the manuscript been provided?**

Reviewer #1: Yes

Reviewer #2: None

Reviewer #3: Yes

PLOS authors have the option to publish the peer review history of their article (what does this mean?). If published, this will include your full peer review and any attached files.

Reviewer #1: **Yes: **Thomas C.T. Michaels

Reviewer #2: No

Reviewer #3: No

Figure Files:

Data Requirements:

Reproducibility:

References:

---

## [Decision Letter · Decision Letter 1]

13 Apr 2021

Dear Dr Tournus,

We are pleased to inform you that your manuscript 'Insights into the dynamic trajectories of protein filament division

revealed by numerical investigation into the mathematical model

of pure fragmentation' has been provisionally accepted for publication in PLOS Computational Biology.

Best regards,

Philip K Maini

Associate Editor

PLOS Computational Biology

Mark Alber

Deputy Editor

PLOS Computational Biology

Reviewer's Responses to Questions

**Comments to the Authors:**

Reviewer #1: The authors have satisfactorily addressed my comments. I therefore recommend publication of this article.

Reviewer #3: The authors have done an excellent job at revising the manuscript, I recommend to accept for publication.

**Have the authors made all data and (if applicable) computational code underlying the findings in their manuscript fully available?**

Reviewer #1: Yes

Reviewer #3: Yes

PLOS authors have the option to publish the peer review history of their article (what does this mean?). If published, this will include your full peer review and any attached files.

Reviewer #1: **Yes: **Thomas Michaels

Reviewer #3: No

---

## [Editor Report · Acceptance letter]

26 Aug 2021

PCOMPBIOL-D-21-00019R1 

Insights into the dynamic trajectories of protein filament division
revealed by numerical investigation into the mathematical model
of pure fragmentation

Dear Dr Tournus,

I am pleased to inform you that your manuscript has been formally accepted for publication in PLOS Computational Biology. Your manuscript is now with our production department and you will be notified of the publication date in due course.

With kind regards,

Andrea Szabo
